# Proteomic Analysis of Rap1A GTPase Signaling-Deficient C57BL/6 Mouse Pancreas and Functional Studies Identify an Essential Role of Rap1A in Pancreas Physiology

**DOI:** 10.3390/ijms25158013

**Published:** 2024-07-23

**Authors:** Durrey Shahwar, Sadaf Baqai, Faisal Khan, M. Israr Khan, Shafaq Javaid, Abdul Hameed, Aisha Raza, Sadaf Saleem Uddin, Hina Hazrat, M. Hafizur Rahman, Syed Ghulam Musharraf, Maqsood A. Chotani

**Affiliations:** 1Molecular Signaling Laboratory, Dr. Panjwani Center for Molecular Medicine and Drug Research (PCMD), International Center for Chemical and Biological Sciences, University of Karachi, Karachi 75270, Pakistan; durreshahwar305@gmail.com (D.S.); sadaf_baqai@hotmail.com (S.B.); shafaq_91@live.com (S.J.); aisha.raza993@gmail.com (A.R.); risha.shaikh@gmail.com (S.S.U.); hina.hrk@outlook.com (H.H.); 2Mass Spectrometry Laboratory, Dr. Panjwani Center for Molecular Medicine and Drug Research, International Center for Chemical and Biological Sciences, University of Karachi, Karachi 75270, Pakistan; khan.faisal@iccs.edu (F.K.); musharraf@iccs.edu (S.G.M.); 3Husein Ebrahim Jamal (H.E.J.) Research Institute of Chemistry, International Center for Chemical and Biological Sciences, University of Karachi, Karachi 75270, Pakistan; 4Molecular Diabetology Laboratory, Dr. Panjwani Center for Molecular Medicine and Drug Research, International Center for Chemical and Biological Sciences, University of Karachi, Karachi 75270, Pakistan; israrkhankpk89@yahoo.com (M.I.K.); hafizpcmd@yahoo.com (M.H.R.); 5Ziauddin College of Molecular Medicine, Ziauddin University, Clifton, Karachi 75600, Pakistan; abdul.hameed2@zu.edu.pk; 6Daffodil International University, Birulia, Savar, Dhaka 1216, Bangladesh; 7Dhaka International University, Satarkul, Badda, Dhaka 1212, Bangladesh

**Keywords:** pancreas, intracellular signaling, Rap1A GTPase, gene knock-out, differential expression, nanoLC-ESIMS/MS, Ero1lβ expression, oral glucose tolerance test, insulin tolerance test, glucose-stimulated insulin secretion

## Abstract

Ras-related Rap1A GTPase is implicated in pancreas β-cell insulin secretion and is stimulated by the cAMP sensor Epac2, a guanine exchange factor and activator of Rap1 GTPase. In this study, we examined the differential proteomic profiles of pancreata from C57BL/6 Rap1A-deficient (Null) and control wild-type (WT) mice with nanoLC-ESI-MS/MS to assess targets of Rap1A potentially involved in insulin regulation. We identified 77 overlapping identifier proteins in both groups, with 8 distinct identifier proteins in Null versus 56 distinct identifier proteins in WT mice pancreata. Functional enrichment analysis showed four of the eight Null unique proteins, ERO1-like protein β (Ero1lβ), triosephosphate isomerase (TP1), 14-3-3 protein γ, and kallikrein-1, were exclusively involved in insulin biogenesis, with roles in insulin metabolism. Specifically, the mRNA expression of Ero1lβ and TP1 was significantly (*p* < 0.05) increased in Null versus WT pancreata. Rap1A deficiency significantly affected glucose tolerance during the first 15–30 min of glucose challenge but showed no impact on insulin sensitivity. Ex vivo glucose-stimulated insulin secretion (GSIS) studies on isolated Null islets showed significantly impaired GSIS. Furthermore, in GSIS-impaired islets, the cAMP-Epac2-Rap1A pathway was significantly compromised compared to the WT. Altogether, these studies underscore an essential role of Rap1A GTPase in pancreas physiological function.

## 1. Introduction

Rap1A belongs to the Ras superfamily of small monomeric G-proteins (Ras-proximate proteins) that participate in signal transduction pathways [1]. Rap1 signaling is regulated by Rap-GEFs (guanine nucleotide exchange factors), which are activated by the highly motile second messengers cAMP, DAG, and IP3, which transform Rap1 from an inactive GDP-bound conformation to an active GTP-bound conformation. Rap1 GTPases are a “molecular switch” because their basic biochemical enzymatic activity or the transition between the two states is controlled by other regulating proteins, Rap-GEFs (guanine exchange factors), and Rap-GAPs (GTPase-activating proteins), which stimulate GTP binding and hydrolysis to form active GTPase and then return it back into its inactive state [2,3]. 

Rap1 GTPases have many known physiological roles, including roles in hematopoietic tissues in which they control adhesion, migration (macrophage, B-cells, and mesenchymal stem cells), superoxide production (neutrophils), and aggregation (platelets). Rap1 signaling is also involved in cell–cell contact and gap junction formation in cardiac myocytes [3,4]. Furthermore, the role of Rap1A GTPase in the expression and trafficking of microvascular α_2C_-adrenoceptors contributing to blood vessel constriction has been extensively studied [5,6,7,8]. 

Moreover, in the pancreas, Rap1 plays a significant role in the cAMP-dependent stimulation of insulin secretion from islet cells. Studies utilizing recombinant adenovirus-transduced cultured rat pancreatic islets support a role for Rap1 subtype A (Rap1A) in insulin release [9]. Further, cAMP sensor and guanine exchange factor Epac2/Rap1 signaling are required for cAMP-mediated regulation of insulin granule dynamics [10]. Other studies revealed that Epac2 binds directly to secretory granule-associated SNARE proteins that are involved in the regulation of insulin granule exocytosis [11,12]. From the data reviewed, it is apparent that Rap1A may play a significant role in insulin exocytosis as Epac2 activates Rap1A signaling in the β-cells of the pancreas. However, the in vivo role played by Rap1A in insulin regulation has not been explored, and the identity of the proteins behind those mechanisms remains unknown. For that purpose, we aimed to investigate the pancreatic proteomics of *Rap1A* knock-out ^(−/−)^ mice versus wild-type ^(+/+)^ mice in order to discover the distinct pattern of protein expression profiling between the two experimental groups by executing a gel-based proteomic approach coupled to nanoscale LC-ESI-MS/MS. We have further studied the differentially expressed proteins with detectable fold changes between the two states of knock-outs ^(−/−)^ and wild types ^(+/+)^ by using bioinformatics for the over-representative analysis of identifier proteins that are closely linked to pathways involved in insulin metabolism, trafficking, and biogenesis. The outcomes of this approach were validated by assessing relative gene expression of select candidates by qPCR and by Western blot analysis.

To complement these studies, we have also carried out physiological characterization of Rap1A-deficient mice in terms of glucose responsiveness and insulin secretion, which further supports the outcomes of proteomic analysis. 

## 2. Results

### 2.1. Proteomic Analysis of Rap1A-Deficient Pancreas

#### 2.1.1. Genotyping

The genomic DNA was isolated from experimental group mouse tail snips (wild type ^(+/+)^ and *Rap1A* knock-out ^(−/−)^) and used in PCR reaction for genotyping. Rap1A wild-type ^(+/+)^ mice produced a PCR product of 1358 bp, while *Rap1A* knock-out ^(−/−)^ mice produced a PCR product of 1107 bp (Appendix A). The *Rap1A* heterozygous ^(−/+)^ mice produced PCR products of both 1358 bp and 1107 bp. The PCR amplicons were identified by agarose gel electrophoresis and a UV-illuminated gel densitometer. The PCR-amplified 1358 bp product indicated intact exon 4 of Rap1A, and the 1107 bp product indicated disrupted exon 4 of Rap1A with insertion of the neo gene [7,13]. 

Based on the outcomes of genotyping, therefore, groups of mice were identified for the studies described below. To increase the likelihood of detecting differential protein expression and effects on physiological function, we utilized mice or tissue from wild-type ^(+/+)^ (naturally expressing Rap1A) and knock-out ^(−/−)^ (deficient for Rap1A) mice to directly compare differences. However, for immunohistochemistry, immunofluorescence, and ex vivo GSIS studies, we included heterozygous ^(−/+)^ (partial loss of Rap1A) mice to assess the effect of graded Rap1A expression on tissue islet morphology and on insulin secretion in the absence or presence of pharmacological inhibitors.

#### 2.1.2. Pooled Sample SDS-PAGE for Detection of Differential Bands

To make sure that there would be less variation in the samples through which we could see a differential protein band pattern on SDS-PAGE, we pooled pancreatic tissue lysates for five wild-type ^(+/+)^ and for five *Rap1A* knock-out ^(−/−)^ samples (in triplicate). The gel image was acquired from the software Quantity One® (version 4.6.3, BioRad Laboratories, Hercules, CA, USA) after staining with Coomassie Blue. This allowed visual detection and assessment of expression patterns between the two genotypes (Figure 1).

#### 2.1.3. Densitometry Analysis of SDS-PAGE (1D Gel)

Due to the absence of bands and differences in expression between wild-type and *Rap1A* knock-out ^(−/−)^ samples, densitometry analysis of SDS-PAGE was performed using Image Lab Software (version 6.1, BioRad Laboratories), which represents the band intensities or optical densities of selected protein bands (Figure 2).

This approach, in particular, visually showed the main differences between the two genotypes prior to nanoLC-ESI-MS/MS.

#### 2.1.4. Semi-Quantification and Expression Analysis of Selected Protein Bands

The amount of signal intensity of each band in both experimental groups was expressed as band percentage (%) in densitometry analysis (Figure 3) and their mean values were used to calculate log2 fold change to check whether the expression levels of whole pancreatic tissue proteins in each band were up-regulated or down-regulated (Appendix A).

#### 2.1.5. Identification of Differentially Expressed Proteins by NanoLC-ESI-MS/MS

The peptide extracts obtained after tryptic digestion from excised gel bands of both experimental groups were subjected to nanoLC-ESI-MS/MS. Protein scores greater than 56 were considered significant. A protein had to include at least two unique peptides in order to be quantified.

The lists of proteins that were identified as unique in *Rap1A* knock-out ^(−/−)^ and in Rap1A wild-type samples are shown in Appendix A, respectively, whereas the list of overlapping proteins expressed in both wild-type and *Rap1A* knock-out ^(−/−)^ samples is shown in Appendix A. 

Differentially expressed proteins were therefore specifically identified for further analysis. 

#### 2.1.6. Protein–Protein Network Interaction Analysis

The expression profile of the two genotypes is summarized in Figure 4A. The protein–protein interaction network analysis was performed using the STRING database on *Rap1A* knock-out ^(−/−)^ and wild-type samples and proteins that overlapped between the two groups (Figure 4B, Appendix A). There were 8 differentially expressed proteins in the *Rap1A* knock-out ^(−/−)^ group and 57 differentially expressed proteins in the wild-type group. Through the interaction network in the STRING database, we predicted three distinct interactions among the *Rap1A* knock-out ^(−/−)^ group unique identifier proteins, including (1) receptor of activated protein C kinase 1 (RACK1/Gnb2l1) with Vigilin (Hdlbp), (2) 14-3-3 protein gamma (Ywhag) with Tubulin alpha-1B chain (Tuba1b), and (3) triosephosphate isomerase (Tpi) with another node in the network. In wild-type and overlapping protein groups, proteins with the highest degree centrality in the PPI network were selected. With the assistance of Cytoscape, all proteins were ranked according to their degree of centrality using the results from the STRING database. Distinct interactions were therefore uncovered for the two genotypes. 

#### 2.1.7. Gene Ontology *Rap1A* Knock-Out ^(−/−)^ and Wild-Type Groups in Terms of Biological Process

Enrichment indicates the extent to which genes associated with a specific pathway are over-represented. We identified unique identifier proteins in Rap1A knock-out and wild-type samples that were enriched for biological processes (Figure 5A,B). In *Rap1A* knock-out ^(−/−)^ samples, we found five biological processes regarding insulin or CHO metabolism, including insulin metabolic processes, protein processing, protein maturation by protein folding, carbohydrate homeostasis, and glucose homeostasis, in which *Rap1A* knock-out ^(−/−)^ unique identifier proteins are involved (Appendix A).

Furthermore, gene ontology in terms of molecular function and cellular components of *Rap1A* knock-out ^(−/−)^ samples, biological processes, molecular functions, and cellular components of the wild-type group are shown in Appendix A.

Overall, these results support specific roles of Rap1A which are linked to pancreas physiological function and specifically related to insulin.

#### 2.1.8. KEGG Pathway Analysis of Differentially Expressed *Rap1A* Knock-Out ^(−/−)^ and Wild-Type ^(+/+)^ Samples

After analyzing the KEGG pathways of *Rap1A* knock-out ^(−/−)^ and wild-type ^(+/+)^ unique identifier proteins, we discovered that Ero1β, a *Rap1A* knock-out ^(−/−)^ unique identifier protein, is involved in protein processing in the endoplasmic reticulum, and Tpi1, another *Rap1A* knock-out ^(−/−)^ unique identifier protein, is involved in Glycolysis/Gluconeogenesis (Appendix A), while in wild-type ^(+/+)^ samples, the identifier proteins Acyp1, Fh1, and Acat1 are involved in pyruvate metabolism; Canx, Rrbp1, Nsfl1c, and Ddost are involved in protein processing in the endoplasmic reticulum; Fh1 and Suclg2 are involved in the citrate cycle (TCA cycle); and Acat1 and Apoa are involved in fat digestion and absorption (Appendix A).

#### 2.1.9. Wild-Type ^(+/+)^ and *Rap1A* Knock-Out ^(−/−)^ Unique Protein IDs’ Distinct Roles in Regulation of Insulin

In terms of insulin trafficking, metabolism, and secretion, there are specific proteins exclusively present in wild-type ^(+/+)^ and *Rap1A* knock-out ^(−/−)^ samples, and we discovered that they play a role in insulin specification/biogenesis. The wild-type proteins and their distinct roles in insulin secretion, pathway linkages, trafficking, and insulin metabolism were investigated using a pie chart (with reference to Panther Database) that shows all wild-type identifier proteins contributed to 42 different pathways in which we discovered six sections that were related to pancreatic secretion or play an important function in insulin metabolism (Figure 6A).

In comparison to the wild type, the unique identifier proteins found exclusively in *Rap1A* knock-out ^(−/−)^ mice samples also have an effect on insulin biogenesis. Four of the eight proteins that are linked to *Rap1A* knock-out ^(−/−)^ mice have been shown to have an important effect/role in insulin biogenesis. Glycolysis, insulin processing, GSIS-mediated insulin secretion, and insulin-induced GLUT4 translocation are all aided by Tpi1, ERO1, 14-3-3 protein gamma, and Kallikrein-1, as shown in Figure 6B, in which the *Rap1A* knock-out ^(−/−)^ mice pancreatic sample has two portions of 0.4% and 0.7% from the remaining portion of the chart belonging to insulin signaling.

For validation, we assessed mRNA expression of ERO1-like protein β (Ero1lβ) and triosephosphate isomerase (Tpi1) using qPCR, which showed significantly higher expression in *Rap1A* knock-out ^(−/−)^ versus wild-type ^(+/+)^ samples (*p* < 0.05, Figure 7A,B).

Specifically, we further assessed protein levels of ERO1lβ (expressed as ratio of ERO1lβ to β-actin signal intensities), which were significantly increased (*p* < 0.01, 1.5-fold) in *Rap1A* knock-out ^(−/−)^ (0.01945 ± 0.0014) versus wild-type ^(+/+)^ (0.01294 ± 0.0007) pancreas samples (Figure 7C,D). The expression of β-actin was similar in both genotypes (*p* = ns, Figure 7E).

These results therefore support the outcomes of the proteomic analysis and identify altered expression of ERO1lβ in pancreas of Rap1A-deficient mice.

### 2.2. Physiological Assessment of Rap1A-Deficient Mice

#### 2.2.1. CBC and HbA1c

A complete blood count (CBC) was performed to assess the overall health of *Rap1A* knock-out ^(−/−)^ versus wild-type ^(+/+)^ mice, particularly for evidence of anemia and effect on hemoglobin level [15]. The results are tabulated in Table 1 and represented as mean ± SEM. There was no significant difference between the two groups in all parameters of the complete blood count (*p* > 0.5).

We further assessed glycated hemoglobin (HbAIc), which is a measure of the average blood sugar levels over the past three months and is used to diagnose prediabetes and diabetes in humans [16] and similarly in mouse models of diabetes [17,18]. However, HbA1c levels were similar in *Rap1A* wild-type ^(+/+)^ (WT, 4.067 ± 0.1382) and knock-out ^(−/−)^ mice (4.10 ± 0.1183, *p >* 0.5*)*. The results are represented as mean ± SEM (Figure 8).

These results show no apparent effect on CBC and HbA1c in Rap1A-deficient mice. 

#### 2.2.2. Oral Glucose Tolerance and Insulin Tolerance Tests


*Altered response to glucose challenge in Rap1A knock-out ^(−/−)^ mice*


To assess how well mice can process large amounts of sugar, experiments were performed to assess the relative duration of blood glucose clearance in *Rap1A* wild-type ^(+/+)^ and knock-out ^(−/−)^ mice using a glucose tolerance test. This test showed that blood glucose peaked at 15 min in both groups. However, in Rap1A-deficient animals, blood glucose levels (346.063 ± 16.167 mg/dL; *p <* 0.001) were remarkably higher than Rap1A control mice (232.75 ± 12.95 mg/dL) (Figure 9A). A similar trend was observed at 30 min; blood glucose levels in Rap1A-deficient mice (247 ± 21.952 mg/dL; *p <* 0.01) remained increased compared to wild-type mice (179.0 ± 13.76 mg/dL). The level of insulin during fasting (assessed at 0 min) was relatively reduced in Rap1A-deficient mice (0.316 ± 0.026 ng/mL; *p* > 0.05) compared with wild-type mice (0.447 ± 0.102 ng/mL). After 15 min of glucose administration, insulin peaked in both experimental groups; this coincided temporarily with the glucose peak. Insulin levels in Rap1A-deficient mice (0.695 ± 0.094 ng/mL) were reduced compared to the control group (0.925 ± 0.117 ng/mL; *p* > 0.05) (Figure 9B). Next, we assessed sensitivity to exogenous insulin in both genotypes. In this insulin tolerance test, all readings after insulin challenge, showed that blood glucose levels were reduced continuously more or less the same in both genotypes (*p* = ns) (Figure 9C).

These data show altered response to glucose challenge in Rap1A-deficient mice and suggest that Rap1 subtype A likely affects insulinotropic potential compared to insulin sensitivity.

#### 2.2.3. Insulin Secretion from Isolated Islets


*Rap1A signal transduction is required for insulin secretion from murine C57BL/6 isolated pancreatic islets after higher glucose challenge*


Pancreas histology using H&E and insulin/DAPI co-stained sections overall showed no gross alterations in Rap1A-deficient islets (Figure 10A,B). 

The measurement of islet area in H&E stained sections showed non-significant differences (*p* = ns) with wild-type ^(+/+)^ area (14,753.88 ± 2372.86 μm^2^, n = 43 islets), *Rap1A* heterozygous ^(−/+)^ area (13,897.98 ± 2233.76 μm^2^, n = 43 islets), and *Rap1A* knock-out ^(−/−)^ area (13,913.45 ± 1792.3 μm^2^, n = 60 islets) (Appendix A). 

Similarly, the measurement of the total β-cell area of anti-insulin-stained section images showed non-significant differences (*p* = ns) with wild-type ^(+/+)^ area (12,332.44 ± 1721.53 μm^2^, n = 32 islets), *Rap1A* heterozygous ^(−/+)^ area (12,907.97 ± 1960.71 μm^2^, n = 28 islets), and *Rap1A* knock-out ^(−/−)^ area (13,094.94 ± 2124.79 μm^2^, n = 27 islets) (Appendix A).

These pancreatic islets were isolated for ex vivo studies to specifically assess the role of Rap1A signaling in insulin secretion.

In the initial experiments, we assessed insulin secretion from islets isolated from *Rap1A* wild-type ^(+/+)^ and heterozygous ^(−/+)^ C57BL/6 mice using 3-isobutyl-1-methylxanthine (IBMX, 0.1 mM), an inhibitor of cyclic nucleotide phosphodiesterases which can increase intracellular levels of cAMP and, along with a stimulatory glucose dose of 16.7 mM, can potentiate insulin release [19]. Size-matched islets were pooled into two different sets. Set one was incubated with 16.7 mM glucose for 1 h/37 °C, while the remaining set was incubated with 16.7 mM glucose plus IBMX for 1 h/37 °C. 

These studies showed that the release of insulin was reduced in islets derived from *Rap1A* heterozygous ^(−/+)^ islets (11.55 ± 3.0 ng/islet/hr; *p* > 0.05) compared with islets from *Rap1A* wild-type ^(+/+)^ mice (15.96 ± 1.48 ng/islet/hr; n = 2 independent replicates, with two islets/replicate) with stimulatory glucose alone. However, in islets incubated with stimulatory glucose concentration plus IBMX, the insulin release was elevated approximately 10-fold in both *Rap1A* wild-type ^(+/+)^ and heterozygous ^(−/+)^ islets. Interestingly, in the presence of IBMX, insulin release was significantly reduced in *Rap1A* heterozygous ^(−/+)^ islets (114.96 ± 9.86 ng/islet/hr; *p* < 0.05) compared with that of the wild-type ^(+/+)^ islets (178.4 ± 14.89 ng/islet/hr; n = 2 independent replicates, with two islets/replicate) (Appendix A). Similarly, in a separate study, we incubated *Rap1A* wild-type ^(+/+)^ and heterozygous ^(−/+)^ islets with the adenylyl cyclase activator and cAMP-elevating agent forskolin, along with glucose (16.7 mM), and found reduced insulin secretion in *Rap1A* heterozygous islets ^(−/+)^ (79.695 ng/islet/hr) compared to wild-type ^(+/+)^ islets (218.645 ng/islet/hr; n = 1, with two islets/replicate) (Appendix A).

Together, these initial outcomes supported the role of cAMP-Epac-Rap1A signaling in insulin secretion. 

Further experiments were therefore performed on islets isolated from *Rap1A* wild-type ^(+/+)^, heterozygous ^(−/+)^, and knock-out ^(−/−)^ mice to assess the impact of Rap1A deficiency on insulin secretion on isolated pancreatic islets supplemented with 3 mM (low, non-stimulatory) or 17 mM (high, stimulatory) glucose [20,21] in the absence or presence of selective protein kinase A (PKA) and Epac2 inhibitors (H-89 and MAY-0132), respectively. For these studies, we assessed islet insulin content for differences in expression level due to Rap1A deficiency. Insulin release was therefore further assessed in reference to insulin content. 

Under similar conditions, the baseline insulin levels were insignificantly reduced in *Rap1A* heterozygous ^(−/+)^ (0.823 ± 0.122% insulin content; *p* > 0.05) and knock-out ^(−/−)^ islets (0.803 ± 0.071% insulin content; *p* > 0.05) versus wild-type ^(+/+)^ islets (1.061 ± 0.16% insulin content) (Figure 10C). Further, comparable insulin responses were exhibited by *Rap1A* wild-type, heterozygous, and knock-out mice in the 3 mM basal, non-stimulatory glucose condition (1.04 ± 0.21, 1.29 ± 0.11, and 0.85 ± 0.05% insulin content, respectively). In the 17 mM stimulatory glucose condition, there was significant increase in insulin secretion in Rap1A wild-type ^(+/+)^ (5.44 ± 0.82% insulin content, *p* < 0.01), heterozygous ^(−/+)^ (5.07 ± 0.49% insulin content, *p* < 0.01), and knock-out ^(−/−)^ (2.46 ± 0.15% insulin content, *p* < 0.001) islets compared to the respective 3 mM basal glucose condition. In the 17 mM stimulatory glucose condition, there was an insignificant difference in insulin secretion in *Rap1A* heterozygous (5.07 ± 0.49% content; *p =* ns) islets compared to *Rap1A* wild-type (5.44 ± 0.82% content) islets. However, in the 17 mM stimulatory glucose condition, insulin secretion was significantly reduced in Rap1A-deficient (2.46 ± 0.15% content; *p <* 0.05) islets compared to Rap1A wild-type (5.44 ± 0.82% content) islets. 

The islets treated with only 17 mM glucose served as control for islets treated with the pharmacological inhibitors H-89 (inhibitor for PKA) and MAY-0132 (selective for Epac2). Inhibition with H-89 slightly lowered glucose-stimulated insulin stimulation (GSIS) in *Rap1A* wild type ^(+/+)^ (4.2 ± 0.70% insulin content; *p =* ns) compared to control, whereas in *Rap1A* heterozygous ^(−/+)^ and knock-out ^(−/−)^ islets, insulin secretion was significantly decreased in the presence of H-89 (2.45 ± 0.48% content; *p <* 0.05 and 0.83 ± 0.05% content; *p <* 0.001, respectively) compared to their controls. Contrary to the effect of the PKA inhibition, the inhibition of Epac2 with MAY-0132 significantly affected GSIS in *Rap1A* wild-type (2.46 ± 0.39; *p* < 0.05) compared to wild-type control islets, while the GSIS in *Rap1A* heterozygous (3.79 ± 0.23% content; *p =* ns) and knock-out islets (1.96 ± 0.1% content; *p =* ns) was less affected in the presence of MAY-0132 (Figure 10D). 

Together, these studies support a major role of Rap1A signaling in pancreas physiology, including a role in regulating insulin secretion.

## 3. Discussion

GSIS (glucose-stimulated insulin secretion) is the process by which pancreatic islet cells release insulin in response to elevated blood glucose levels, especially after meals [22]. Insulin stimulates glucose uptake in primary tissues like skeletal muscle and adipose tissue while inhibiting liver gluconeogenesis [23]. When β-cells detect glucose, they absorb it through glucose transporters (GLUT1 in humans and GLUT2 in rodents) and begin glucose metabolism; this physiological response is triggered at high glucose concentrations and not at low glucose concentrations and therefore prevents natural hypoglycemia [24,25]. Glucose metabolism causes a chain reaction of signaling events that changes the ATP/ADP ratio, closes ATP-sensitive potassium channels, and depolarizes the plasma membrane (PM). These actions activate voltage-dependent Ca^2+^ channels (VDCCs) in the PM, and the influx of Ca^2+^ from the extracellular space causes the β-cells to release insulin quickly from pre-packaged insulin granules [26,27]. The insulin secretory process is divided into two parts: the first, which lasts about 10 min, is when a large amount of insulin is released quickly. The second part is when less insulin is released, but it only lasts as long as there is high glucose in the blood [28].

Specifically, the rise in ATP elevates intracellular cAMP levels promoting insulin secretion by two different pathways; the protein kinase A-dependent pathway and an Epac2A/Rap1 signaling pathway. Epac2 activates Rap1 when it is bound with cAMP [29,30]. Epac2/Rap1 are the most abundant members of their families in islet cells [31,32]. The A-kinase phosphorylation of Rap1-GAP prevents Rap-GAP from hydrolyzing Rap1 GTP, thereby maintaining Rap1 activity [33]. Activation of Rap1 causes activation of phospholipase C-epsilon (PLC-ε). PLC-ε initiates the synthesis of diacylglycerol (DAG) and inositol triphosphate (IP3) and subsequent activation of Protein kinase C (PKC) and Ca^2+^ release. The PKC phosphorylates secretory granule-associated proteins, increasing the likelihood of insulin granule breakdown. 

Indeed, studies using Epac2 knock-out mice demonstrated that this Rap GEF is crucial for insulin granular exocytosis, primarily the cAMP-mediated early phase of insulin release, by increasing the “restless newcomers”, defined as granules that are newly recruited to the PM upon stimulus and immediately fuse with it [10]. The same study also showed co-localization of Rap1 with insulin granules in cultured mouse insulin-producing MIN6 cells. Upon siRNA-mediated Rap1 knockdown (which included both Rap1A and 1B), the cAMP-mediated insulin secretion was reduced by ~40%. Therefore, Epac2/Rap1 signaling may increase the overall pool size of non-docked granules and/or may play an active part to recruit these granules to the cellular surface [10]. Delayed insulin release, particularly from restless newcomers, may therefore occur in the absence of Rap1.

Therefore, with reference to the role of Rap1 GTPases in insulin secretion and the mechanism of insulin secretion, analysis of nanoLC-MS/MS mass-spectrometry data from the pancreas of experimental groups identified 56 protein IDs unique to *Rap1A* wild-type ^(+/+)^, 15 of which were specifically involved in insulin secretion, and 8 protein IDs that were unique to *Rap1A* knock-out ^(−/−)^. The wild-type protein identifiers that share GSIS-dependent insulin secretion are as follows: FUMH: fumarate hydratase, mitochondrial (Fh1); APOA4: Apolipoprotein A-IV (ApoA4); TM263: Transthyretin (Ttr); NUCB2: Nucleobindin-2.

We have also classified a number of protein identifiers into various insulin-related functions. LMNA: Prelamin-A/C (Lmna) is an identifier protein that has been shown to play a role in type 1 diabetes (T1DM). Additionally, we identified Rap1A protein IDs associated with inflammation, as inflammation has been shown to cause damage to pancreatic β-cells, including thioredoxin-dependent peroxide reductase, mitochondrial (Prdx3), and GP2: Pancreatic secretory granule membrane major glycoprotein GP2 (Gp2). EWS: RNA-binding protein, Ezrin (Ezr) protein, TMED (Transmembrane emp24-domain-containing protein 10) protein, and CAVN2 (Caveolae-associated protein 2) are directly associated with the endoplasmic reticulum (specifically for insulin biogenesis), transport cargo protein forward, participate in insulin trafficking, and dock at the plasma membrane. Calnexin is a protein identifier that is involved in the modification of the insulin receptor (Canx). The following identifier proteins are involved in insulin secretion and are associated with additional insulin signaling pathways: TCTP (translationally controlled tumor protein-Tpt1), YBOX3 (Y-box-binding protein 3-Ybx3), and TAGL2 (transgelin-2 Tagln2). DCTN2 (Dynactin subunit 2) is a protein that has been linked to cancer and was identified in the pancreata of wild-type mice.

On the other hand, in the *Rap1A* knock-out ^(−/−)^ group, there were eight identifier proteins uniquely expressed, of which four have an impact on insulin metabolism, including endoplasmic reticulum oxidoreductin 1-like protein β (ERO1lβ), an important player in biogenesis of insulin and homeostasis of glucose [14]. ERO1lβ is a key player in disulfide bond formation in the endoplasmic reticulum (ER) and maintains an oxidative environment in the ER to facilitate the disulfide bond [34]. ERO1lβ levels are induced by high glucose concentrations, indicating an increased load of oxidative protein folding in the ER of β-cells. These findings illustrate the role of ERO1 in maintaining insulin content and regulating cell survival during ER stress. Glucose revives insulin biosynthesis, secretion, and transcription. Pre-proinsulin is translated and then imported into the ER, where it is cleaved into proinsulin. The proinsulin forms three disulfide bonds in the oxidative territory of the ER. Subsequently, proinsulin is further cleaved into insulin and C-peptide by pro-hormone convertases. However, under constant glucose stimulation, β-cells increase the insulin biosynthesis and transcription. This increases the demand on the β-cell ER, which ultimately delays proinsulin folding to its native state and minimizes insulin content, leading to decreased or less insulin secretion. Improper protein folding may therefore lead to ER stress and, when it is prolonged, may cause cell death [35,36].

Following a review of evidence-based oxidative stress in ER-related studies on ERO1, we hypothesize that the increased expression of ERO1lβ may be part of the unfolded response (UPR, [36]) in the Rap1A knock-out mouse ^(−/−)^ pancreas when compared to the wild-type pancreas and indicates the insulin-processing load in the ER, coupled with oxidative stress, could lead to delay in insulin processing (proinsulin folding, cleavage to active insulin formation) upon glucose stimulation. To assess this possibility, the levels of proinsulin in the islets of all three genotypes can therefore be examined, which remains the focus of future studies.

Indeed, physiological assessment of these mice showed that at 15 min following glucose administration, blood glucose levels in Rap1A-deficient mice were significantly elevated, and delayed normalization of blood glucose levels versus wild-type mice was apparent. These results suggest postprandial hyperglycemia and thus an effect on glucose tolerance with Rap1A depletion. Moreover, isolated pancreatic islets from *Rap1A*-deficient mice also showed significantly decreased total insulin secretion at the challenging glucose concentration (discussed below).

Upon exposure to potent stimulators such as glucose, β-cells secrete insulin via the cAMP/protein kinase A/Epac2 pathway [37,38,39,40]. To assess whether Rap1A absence affected protein kinase A- and cAMP-mediated insulin secretion ex vivo, we treated islets isolated from *Rap1A* wild-type, heterozygous, and knock-out mice with H-89, a pharmacological inhibitor of protein kinase A, and MAY-0132, a pharmacological inhibitor of Epac2. From earlier studies, it is evident that treatment of pancreatic islets with inhibitors of these two targets causes a marked reduction in glucose-stimulated insulin levels coupled to cAMP signaling-associated pathways, namely cAMP/protein kinase A or cAMP/Epac2 [41,42]. In agreement with previous data, our studies carried out on pancreatic islets supplemented with 17 mM glucose demonstrated that H-89 caused significant inhibition of GSIS on *Rap1A*-deficient and heterozygous ^(−/+)^ islets of Langerhans compared to *Rap1A* wild-type ^(+/+)^ islets. 

In contrast with the protein kinase A inhibitor, MAY-0132, the Epac2 inhibitor, caused significant reduction in glucose-stimulated insulin secretion in *Rap1A* wild-type ^(+/+)^ islets, whereas in *Rap1A*-deficient and heterozygous ^(−/+)^ islets, the MAY-0132-mediated inhibition was not significant compared with the respective controls. Notably, in the presence of the Epac2 inhibitor (MAY-0132), the *Rap1A* wild type ^(+/+)^ showed more or less similar insulin secretory response as exhibited by *Rap1A* knock-out ^(−/−)^ mice at 17 mM alone, suggesting that Rap1A acts co-operatively with Epac2 for insulin release from β-cells. 

Further, in support of these outcomes, in our experimental conditions using islets from BALB/c strain mice, we generally found that H-89 (protein kinase A inhibition) and MAY-0132 (Epac2 inhibition), in addition to other inhibitor IBMX (phosphodiesterase inhibition) and FSK (adenylyl cyclase activation), did not affect insulin secretion in basal glucose condition (3 mM; unpublished observations).

Altogether, the insulin release in *Rap1A* wild-type ^(+/+)^ islets is predominantly coupled to Epac-Rap1 versus protein kinase A signaling; however, in *Rap1A* heterozygous ^(−/+)^ and *Rap1A*-deficient islets, although protein kinase A signaling predominates, it is insufficient for insulin release comparable to the release observed in wild-type islets. Combined with the outcomes of preliminary studies utilizing IBMX and forskolin, the Epac2-Rap1A signaling therefore plays a major role in insulin secretion in islets of C57BL/6 strain mice. The differentially expressed proteins identified in this study may be the potential targets of this signaling, which can be further assessed in future studies. 

## 4. Materials and Methods

### 4.1. Animals

Rap1A heterozygous mice were bred to generate mice for the experimental studies. Pups were screened for their genotype by PCR to identify control wild-type ^(+/+)^ and Rap1A knock-out ^(−/−)^ (test group) mice. All mice were ear-tagged with designated numbers. The mean ± SEM age in months of male mice used for the studies was 6.27 ± 0.46. 

The experimental groups included the following:(a)*For proteomic studies*: total of 10 male mice (5 wild type ^(+/+)^ and 5 knock-out ^(−/−)^). Pancreatic tissue from the experimental group was harvested and stored at −80 °C.(b)*For RNA analysis studies*: see Section 4.6.1.(c)*For Western blot analysis*: see Section 4.7.(d)*For physiological studies*: see Section 4.8.

All procedures involving animal handling and use were reviewed and approved by the Institutional Animal Care and Use Committee (IACUC, protocol no. 2018-0004), which were in compliance with international guidelines and standard protocols for animal use in research.

### 4.2. Proteomic Strategy

For the downstream analysis of pancreatic tissue samples for proteomics, proteins were extracted using urea-denaturing cocktail (4 M Urea, 0.2 mM EDTA, 0.8 mM NaCl, 0.4% CHAPS, 0.15 mM DTT, 2 mM Tris-HCl, 1 mM sodium orthovanadate, protease inhibitor) to extract proteins directly from pancreatic tissue samples. The Bicinchoninic Assay (BCA), was used to estimate total protein abundance from the extracted pancreatic tissue lysate. For the separation, crude mixture of estimated proteins was used for the 1-Dimensional SDS-PAGE. Densitometry image analysis (Image Lab Bio-Rad Laboratories, Hercules, CA, USA) of 1D SDS-PAGE was performed to obtain the log fold change for gel bands expression values. 

Samples were pooled (in triplicate). For pooling, the wild-type or Rap1A knock-out samples were mixed together with sample running buffer (2×) in a single tube with 1:1 ratio separately. After mixing, 14 µL (8 µg) from each tube of wild-type or Rap1A knock-out mice samples were loaded in the wells of gel side by side as 1st, 2nd, and 3rd replicates of each experimental group. Differential protein bands from the gel were then excised, followed by in-gel tryptic digestion.

### 4.3. Nanoscale Liquid Chromatography Mass Spectrometry (nanoLC-ESI-MS/MS)

The tryptic digests were dissolved in 0.1% TFA and 1 μL was injected on a C18 reverse-phase trapping column (C18, Acclaim™ PepMap™ 100, 75 μm × 2 cm, particle size 3 μm, nanoViper, Thermo Fisher Scientific, Waltham, MA, USA) and was separated by C18 reverse-phase nano-LC column (75 μm × 15 cm Acclaim™ PepMap™ RSLC C18 column with 2 μm particle size and 100 Å pore size, NanoViper, Thermo Fisher Scientific, Waltham, MA, USA), using Nano-UHPLC, (Dionex, UltiMate 3000 RSLCnano System, Thermo Scientific, Waltham, MA, USA) coupled to MaXis II ESI-QTOF Mass spectrometer (Bruker Daltonics, Bremen, Germany). 

### 4.4. MS/MS Data Analysis

The Bruker Daltonics Compass Data Analysis Software (version 4.4) was used to generate the molecular feature list for each LC-MS/MS run. The features that resulted were loaded into the ProteinScape database management system. ProteinScape was used to identify peptides by searching MS/MS spectra against a Swiss-Prot database with the taxonomy filter *Mus musculus*. Using all of the experiment’s MS/MS spectra, a combined protein list was created. Mascot 2.4 was used to conduct database searches with the following parameters: cysteine carbamidomethylation was set as a fixed modification and methionine oxidation was set as a variable modification. Trypsin, a proteolytic enzyme, was specified, and one missed cleavage was allowed. Peptide mass tolerance was set to 10 ppm, fragment mass tolerance to 15 ppm, and peptide charges to +2, +3, and +4. A Mascot percolator was used to filter the Mascot results, increasing the sensitivity and accuracy of the peptide identification. The significance threshold was *p* < 0.05, and the acceptable rate of peptide false discovery was set at 1%. 

### 4.5. Bioinformatics: Functional Enrichment Analysis of DEPs

To explore the biologically significant results obtained from the nanoLC tandem mass spectrometry raw data, we used bioinformatics tools for gene ontological enrichment analysis (GO terms), protein–protein interaction network analysis (STRING data base), KEGG pathway analysis, and PANTHER database and an online tool, ShinyGo, for the over-representative analysis of differentially expressed identifier proteins.

### 4.6. Gene Expression Analysis

#### 4.6.1. RNA Isolation from Mouse Pancreatic Tissue

Littermate male mice were used for RNA isolations (n = 9 *Rap1A* wild type ^(+/+)^ and n = 9 *Rap1A* knock-out ^(−/−)^) using TRIzol reagent (Thermo Fisher Scientific, Waltham, MA, USA). Extraction of high-quality RNA from ribonuclease-rich tissue like pancreas is challenging as mouse pancreas can contain up to 75 mg ribonuclease [43]. The extraction protocol described by Chougoni and Grossman [44] was modified by storing tissue (~20 mg) in RNA*later*^TM^ stabilization solution (Thermo Fisher Scientific, Waltham, MA, USA) at −80 °C, and before the chloroform step, samples were centrifuged to remove undigested tissue. For validation of intact RNA, agarose gel electrophoresis was performed. Concentration of RNA and absorbance ratio A_260/280_ value were obtained by using Nanodrop^TM^ (Thermo Fisher Scientific, Waltham, MA, USA).

#### 4.6.2. DNase Treatment

Total RNA was treated with DNase I to completely eliminate genomic DNA contamination by using DNase I, RNase-free kit (Thermo Fisher Scientific, Waltham, MA, USA). Briefly, 1 µg RNA was incubated with 1 µL of DNase I and an equal volume of 10× reaction buffer at 37 °C for 45 min. The enzymatic reaction was inactivated by adding 1 µL of 50 mM EDTA and incubating at 65 °C for 10 min. High-speed centrifugation at 12,000 rpm for 1 min separated pure RNA in the supernatant. Concentration of pure RNA was determined by Nanodrop Spectrophotometer. 

#### 4.6.3. cDNA Synthesis

The purified total RNA was used to synthesize complementary DNA (cDNA) using the RevertAid First Strand cDNA synthesis kit ((Thermo Fisher Scientific, Waltham, MA, USA), according to manufacturer instructions. One µg total RNA was incubated with 1 µL oligodT primer and 10 µL of nuclease-free water at 70 °C for 5 min. The cDNA was synthesized by further incubating RNA with 4 µL reaction buffer, 2 µL dNTPs, 1 µL RiboLock RNase Inhibitor, and 1 µL of RevertAid RT at 45 °C for 1 hr, followed by inactivation at 70 °C for 10 min. The cDNA was stored at −80 °C.

#### 4.6.4. Primer Designing

Specific primers against β-Actin *(Actb)*, Endoplasmic Reticulum Oxidoreductase 1 like β *(Ero1lb)*, and triosephosphate isomerase 1 *(Tpi1)* were designed by an online primer designing software (Primer-BLAST (https://www.ncbi.nlm.nih.gov/tools/primer-blast; accessed on 16 January 2023). The NCBI accession numbers and details of primers for each of the three genes are given in Table 2.

#### 4.6.5. Quantitative Real-Time PCR Analysis

The expression profiles of selected genes were analyzed by performing RT-qPCR with BioRad CFX96 system by utilizing the specific primers shown in Table 2. The PCR conditions for each primer pair, including annealing temperatures, were initially optimized, and PCR product of the predicted size was visualized using agarose gel electrophoresis prior to real-time PCR analysis. Briefly, the 10 µL PCR reaction consisted of 2 µL of template cDNA, 1 µL each of forward and reverse primers, 5 µL of BlasTaq^TM^ 2× qPCR MasterMix (Applied Biological Materials Inc. (abm), Richmond, BC, Canada), and 1 µL of nuclease-free water. β-Actin was used as a reference gene and each sample was run in triplicate. The cyclic parameters for PCR consisted of initial denaturation at 95 °C for 10 min and 40 cycles, including denaturation at 95 °C for 30 s, annealing at 57 °C for 1 min, and extension at 70 °C for 1 min. At the end of the reaction, the obtained threshold cycle (Ct) values for each sample were used to determine the fold change (2^−(ΔΔCt)^) in gene expression.

### 4.7. Western Blot Analysis

For protein extraction from *Rap1A* wild-type ^(+/+)^ (n = 5) and *Rap1A* knock-out ^(−/−)^ (n = 5) mice pancreata, 100 mg powdered tissue was lysed in RIPA Lysis and Extraction Buffer (Thermo Fisher Scientific, Waltham, MA, USA) having protease and phosphatase inhibitors (Halt^TM^ Protease and Phosphatase Inhibitor Cocktail-EDTA-free at 1×, Thermo Fisher Scientific, Waltham, MA, USA). The lysate was stored at −20 °C until further processing. The protein concentration was estimated with Bicinchoninic Acid (BCA) assay (Thermo Fisher Scientific, Waltham, MA, USA). Forty µg protein were loaded on SDS-PAGE for both β-actin (5% stacking and 12% resolving gel) and ERO1lβ (5% stacking and 10% resolving gel) detection along with pre-stained protein ladder (Sangon Biotech (Shanghai, China) Co., Ltd., Shanghai, China, TrueColor Pre-stained Protein Marker, 2 colors, Wide Range, 10~250 kDa). After SDS-PAGE complete run, proteins were transferred to polyvinylidene fluoride membrane (PVDF, 0.2 µm, Bio-Rad, Hercules, CA, USA) via electrophoretic run at 90 mV for 50 min in transfer buffer. After protein transfer, membranes were blocked with 5% non-fat dry milk (NFDM) in TBS (Tris-buffered saline, pH 7.4) for 30 min. Both antibodies, anti β-actin (1:15,000 dilution, monoclonal antibody AC-15 (Cat. no. AM-4302), Thermo Fisher Scientific, Waltham, MA, USA) and anti ERO 1-like β (ERO1lβ, 1:2000 dilution, polyclonal antibody (rabbit, Cat. no. PA5-25142), Thermo Fisher Scientific, Waltham, MA, USA) antibodies, were diluted in blocking solution. Blots were incubated in primary antibody overnight at 4 °C. After washing five times with TBST (Tris-buffered saline with Tween 20), again blots were blocked with 5% NFDM for 30 min. Secondary antibodies (1:2000 dilution, goat anti-mouse IgG (H+L) secondary antibody, HRP (Cat. no. 31430) or goat anti-rabbit IgG (H+L), secondary antibody, HRP (Cat. no. 31460), Thermo Fisher Scientific, Waltham, MA, USA) were also diluted in blocking solution. After one-hour incubation in secondary antibodies, membranes were washed with TBST 10 times for five minutes each time. Protein bands were identified with SuperSignal™ West Pico PLUS Chemiluminescent Substrate kit (Thermo Fisher Scientific, Waltham, MA, USA) using the C-DiGit^®^ Blot Scanner system (LI-COR Biosciences, Lincoln, NE, USA). The intensity of band signal was calculated using the Image Studio software of C-DiGit Blot Scanner system (version 5.2.5). The immuno-blotting for both ERO1lβ and β-actin was performed 2–4 times. The ratio of signal intensity for ERO1lβ (mean value of four blots) to signal intensity for β-actin (mean value of two blots) was calculated for each sample. Statistical analysis was performed on GraphPad Prism 8 software using unpaired *t*-test.

### 4.8. Physiological Studies

#### 4.8.1. CBC and HbA1c

The complete blood counts were measured from whole-blood samples using the Beckmann Coulter DxH 800 clinical hematology analyzer (n = 5 for male *Rap1A* wild type ^(+/+)^; n = 6 for male *Rap1A-* knock-out ^(−/−)^).

The HbA1c levels were measured from whole-blood samples using the COBAS c311 Analyzer for HbA1c assay, which is based on photometric transmission measurements (n = 6 male mice for each group). 

#### 4.8.2. Oral Glucose Tolerance Test (OGTT)

Following a 6 h fast, mice were orally administered dextrose, 2g/kg [45]. Blood glucose readings were recorded at 0, 15, 30, 45, 60, and 120 minutes’ after glucose challenge using ACCU-CHEK^®^ Performa glucometer (Roche, Indianapolis, IN, USA). Blood glucose levels of male *Rap1A* wild-type ^(+/+)^ (control; 12 readings per time-point from 4 mice from three independent experiments) and male *Rap1A* knock-out ^(−/−)^ (8 readings per time-point from 6 mice from three independent experiments) mice were measured. The readings were averaged for each mouse prior to analysis.

The serum insulin levels were measured by an ultrasensitive mouse insulin ELISA kit following the manufacturer’s given protocol (Crystal Chem, Elk Grove Village, IL, USA). The samples included male *Rap1A* wild-type ^(+/+)^ (6–7 readings per time-point from 8 mice) and male *Rap1A* knock-out ^(−/−)^ (4–5 readings per time-point from 5 mice) mice at 0, 15, and 30 min of glucose administration.

#### 4.8.3. Insulin Tolerance Test (ITT)

After acclimatization of three days, mice were intraperitoneally injected with insulin (0.5 U/kg, Humulin R, Eli Lilly, Indianapolis, IN, USA) after 6 h of fasting [46], and blood glucose readings noted at 15, 30, 45, and 60 min after insulin dosing were normalized with the baseline (0 min) reading. Two independent experiments were performed on male *Rap1A* wild-type ^(+/+)^ (control; n = 12 readings per time point from 6 mice) and male Rap1A knock-out ^(−/−)^ (n = 9 readings per time point from 6 mice) mice. The readings were averaged for each mouse prior to analysis.

#### 4.8.4. Histology of Pancreata

Male mice of all three groups, *Rap1A* knock-out ^(−/−)^, heterozygous ^(−/+)^, and wild type ^(+/+)^, were dissected, and pancreatic tail was isolated, fixed in acid fixative, processed, and embedded in paraffin. Four-micron sections were cut using a microtome (Yidi, Jinhua, Zhejiang, China) and stained with hematoxylin and eosin (H&E) for overall tissue morphology. Images of stained sections were obtained with Nikon DXM-1200C digital camera (Nikon 90i microscope, Nikon, Tokyo, Japan).

For immunofluorescence, sections were double stained for insulin and nuclei using guinea pig polyclonal anti-insulin primary antibody (Abcam, Cat. no. ab7842, Waltham, MA, USA), secondary Alexa Fluor^®^ 594-AffiniPure F(ab’)_2_ fragment donkey anti-guinea pig IgG (H+L, Jackson ImmunoResearch, Cat no. 706-586-1480, West Grove, PA, USA), and DAPI (Sigma, St. Louis, MO, USA). The fluorescent images were viewed and captured under the 20× objective lens.

For islet area and total β-cell area measurement, tissue sections from three animals of each genotype were assessed and counted at the 27th–32nd layer (section) of tissue. Area of each islet of Langerhans was measured using ImageJ software (version 1.50i) of NIH for H&E-stained sections. Similarly, total β-cell area was measured using anti-insulin-stained section images. Firstly, the units were changed in ImageJ area measurement tool, and the total β-cell area was outlined and measured using freehand tool of ImageJ software (version 1.50i). 

#### 4.8.5. Islet Isolation and Insulin Secretory Activity

Isolation of islets and insulin release assay were performed according to our established protocol [20,21,47]. Briefly, in our experimental setup, we used fresh isolated islets for all the experiments. After isolation and purification, three size-matched fresh islets were pre-incubated for 45 min/37 °C at basal glucose (3 mM) in Kreb’s Ringer Bicarbonate Buffer (KRBB) solution to acclimate the islets to the experimental conditions. Following pre-incubation, the islets were incubated for 60 min/37 °C in KRBB solution with basal glucose concentration (3 mM), glucose of stimulatory concentration (17 mM) alone, and 17 mM glucose supplemented with H-89 (15 µM, Sigma, St. Louis, MO, USA) or with MAY-0132 (10 µM, Sigma, St. Louis, MO, USA), protein kinase A and Epac2 inhibitors, respectively. Subsequently, 100 µL aliquots were collected from each tube after completion of one hour of incubation and quantified by an ultrasensitive mouse insulin ELISA kit (Crystal Chem, Elk Grove Village, IL, USA).

Three independent experiments were performed, each with triplicate batches of size-matched islets/mice and each having 3 islets isolated from 2–3 male mice of each genotype. The triplicate samples of each experiment (1, 2, and 3) were pooled for insulin quantification by ELISA. 


*Estimation of Total Insulin Content*


After the supernatant was taken for measurement of secreted insulin, the KRB media was replaced by same volume of 0.1N HCl. To extract the total content, the islets were vortexed for 30 s on the highest vortex setting. Re-suspended islets were incubated on ice for 2 h with intermittent vortexing every 30 min. To ensure complete lysis, the islets were sonicated with an electric sonicator for 1 s. The sonicated islet extracts were centrifuged at 3,000 rpm for 10 min at 4 °C. The lysates were collected and stored at −40 °C. The total insulin content was measured using ultrasensitive insulin ELISA kit, as described per manufacturer protocol. Insulin content of islets was calculated as sum of secreted insulin and total insulin content. 

Normalization of GSIS was performed with the formula shown below:GSIS insulin content = [secreted insulin ÷ (secreted insulin + total insulin content)] × 100

### 4.9. Statistical Analysis

For statistical significance, one-way ANOVA was applied using GRAPH PAD Prism 8 statistical analysis software (San Diego, CA, USA). Student’s t-test was used to analyze the statistical differences between means of two groups, and two-way ANOVA with Bonferroni post hoc test was applied. Results are shown as mean +/− SEM. *p*-value ≤0.05 was considered as significant. 

## 5. Conclusions

In this study, we investigated the molecular function of Rap1A GTPase using proteomic profiling of pancreatic tissues from wild-type *Rap1A*
^(+/+)^ and *Rap1A* knock-out ^(−/−)^ mice and discovered a distinct pattern of protein expression in both experimental groups. On the basis of the differential bands and missing bands on the gel and differential protein expression using bioinformatics tools, validation by qPCR and Western blot analysis, and functional in vivo and ex vivo studies, we conclude that in wild-type tissue, certain proteins which are required for a proper insulin secretory response were expressed as a result of Rap1A activation. Furthermore, in the knock-out ^(−/−)^ group, Rap1A deficiency may cause the up-regulation of some distinct proteins that play a compensatory role in β-cell insulin processing and folding in the endoplasmic reticulum. Moreover, these findings also lead to an improved understanding of the overall expression of mouse proteomic profiling of the pancreas in the presence and absence of Rap1A GTPase in wild-type ^(+/+)^ and *Rap1A* knock-out ^(−/−)^ mice, which to date has remained unknown.

## Figures and Tables

**Figure 1 ijms-25-08013-f001:**
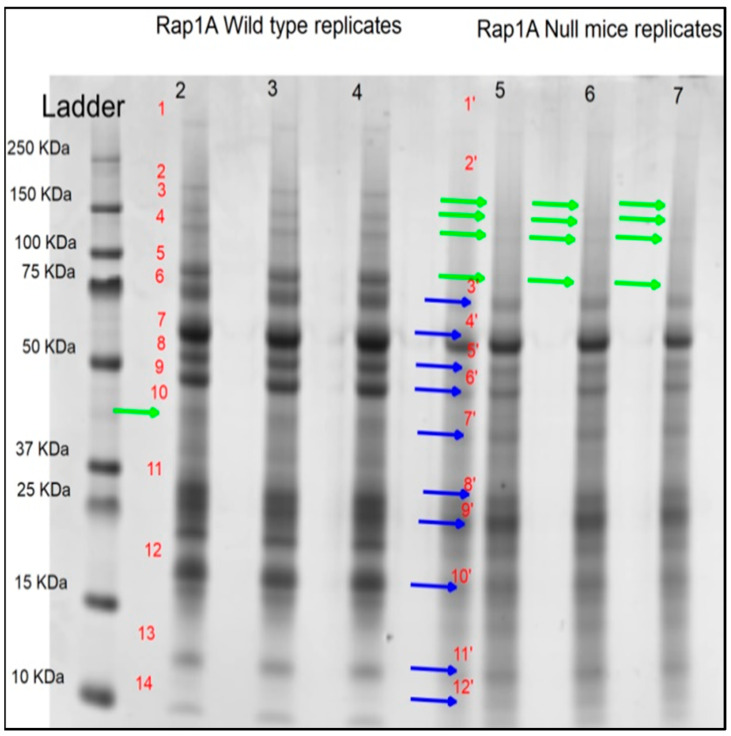
Pooled sample analysis for detection of differential bands in pancreatic tissue samples. The first lane is a protein marker (kDa). The wild-type samples are in lanes 2, 3, and 4, while the *Rap1A* knock-out ^(−/−)^ mouse pancreatic tissue samples are in lanes 5, 6, and 7. The green arrow indicates missing bands; four bands were missing in *Rap1A* knock-out ^(−/−)^ replicates, and a single band was missing in wild-type replicates. The blue arrow indicates that protein expression is differential in both groups of wild-type and *Rap1* knock-out ^(−/−)^ pancreatic tissue. We uncovered missing protein bands and expression differences in *Rap1A* knock-out ^(−/−)^ mouse samples when compared to wild-type samples using sample pooling in SDS-PAGE (numbers 1–14 for wild-type samples, and numbers 1’–12’ for Rap1A-null samples, which were chosen for densitometry analysis).

**Figure 2 ijms-25-08013-f002:**
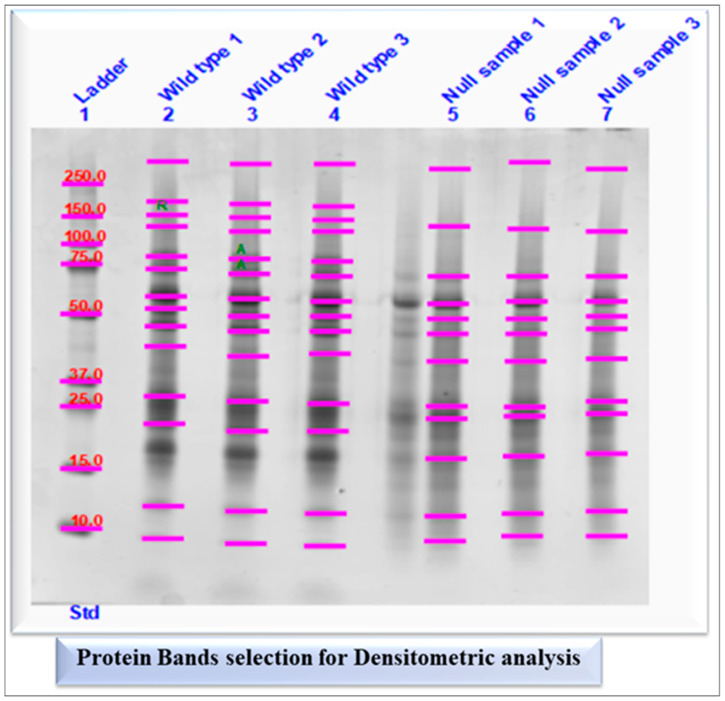
Differential band selection of both experimental groups wild-type and *Rap1A* knock-out ^(−/−)^ (Null) mice samples for densitometry analysis. The letter R denotes the reference band which was used to calculate the relative quantity of the other bands, while the letter A denotes absolute quantification used to quantify bands based on known standard bands using a calibration curve.

**Figure 3 ijms-25-08013-f003:**
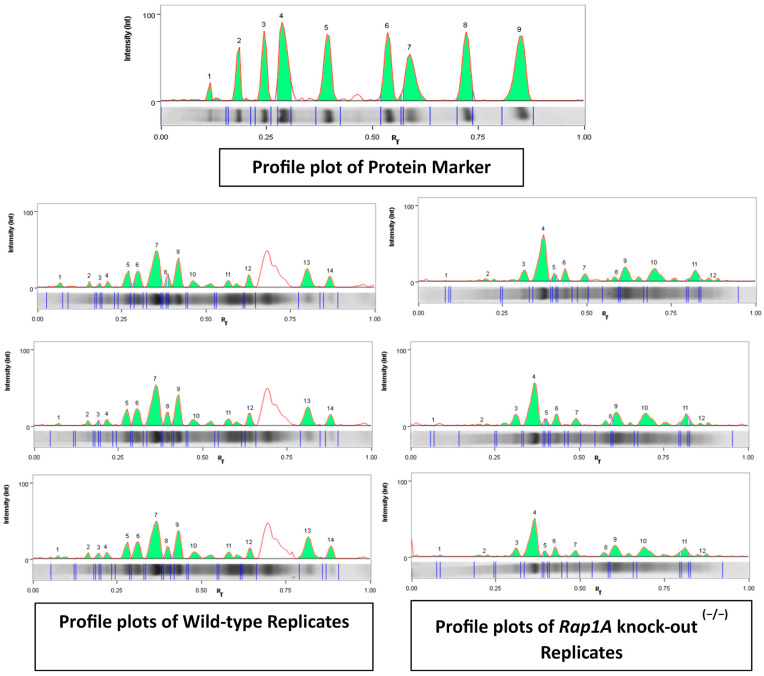
Profile plots of wild-type and *Rap1A* knock-out ^(−/−)^ replicates. The profile plot of each lane shows the signal intensities (green peaks) of differential banding patterns in the pooled sample image from both experimental groups (wild-type and *Rap1A* knock-out ^(−/−)^ samples). Below the lane profile graph, a lane image strip displays each band delimited by a pair of vertical lines. The left panel contained 14 wild-type peaks. There were 12 peaks in the right panel for the *Rap1A* knock-out ^(−/−)^ group. At the top of the panel, there were nine protein biomarker peaks.

**Figure 4 ijms-25-08013-f004:**
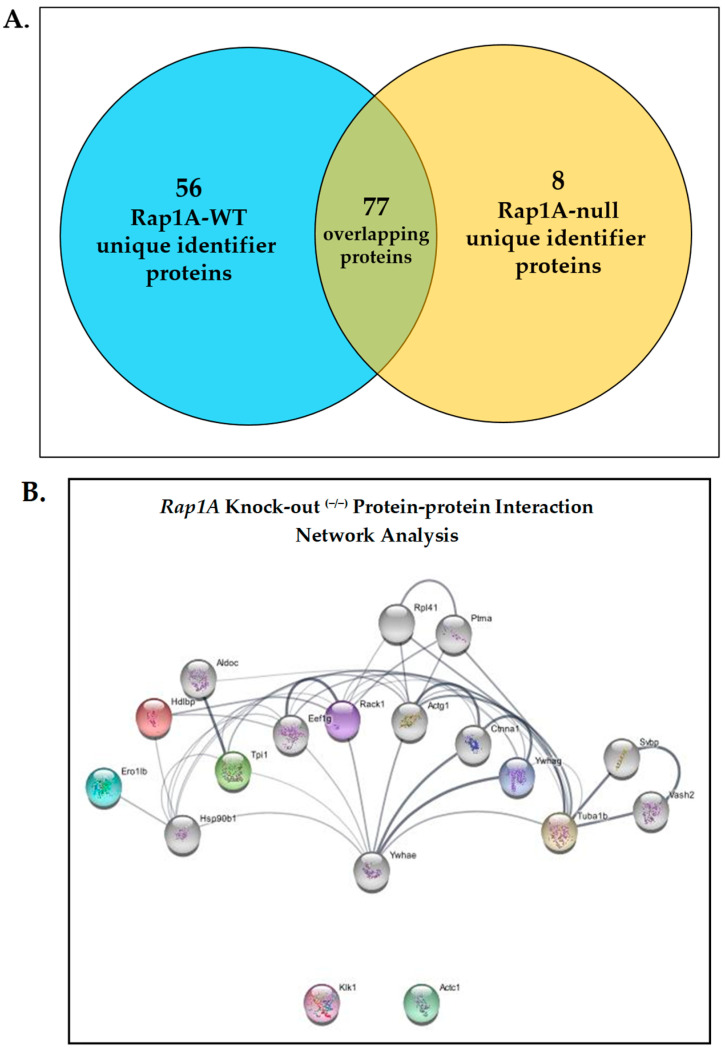
(**A**) A Venn diagram representing the expression profile of the two genotypes. The total number of unique identifier proteins that were expressed in wild type was 56, while *Rap1A* knock-out ^(−/−)^ samples had 8 proteins and 77 overlapping proteins that came from both experimental groups. (**B**) Rap1A knock-out sample PPI network. A total of 18 nodes and 42 PPI network edges with a local clustering coefficient of 0.608.

**Figure 5 ijms-25-08013-f005:**
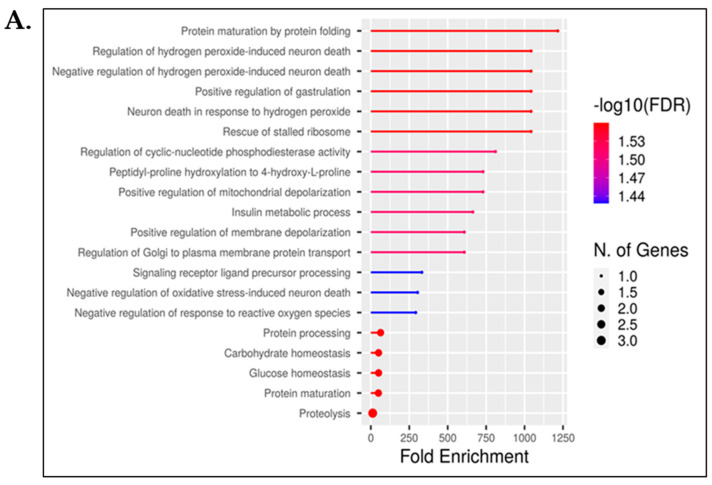
(**A**) Number of *Rap1A* knock-out ^(−/−)^ identifier proteins involved in pathways. The number of genes with their respective pathways are sorted by fold enrichment on the x-axis and by −log10(FDR), which represents the shifts of color intensities changing from high (red) to low (blue). The size of the circles, from small to big, represents the number of genes contributing to the corresponding pathway. (**B**) Number of wild-type identifier proteins involved in biological pathways.

**Figure 6 ijms-25-08013-f006:**
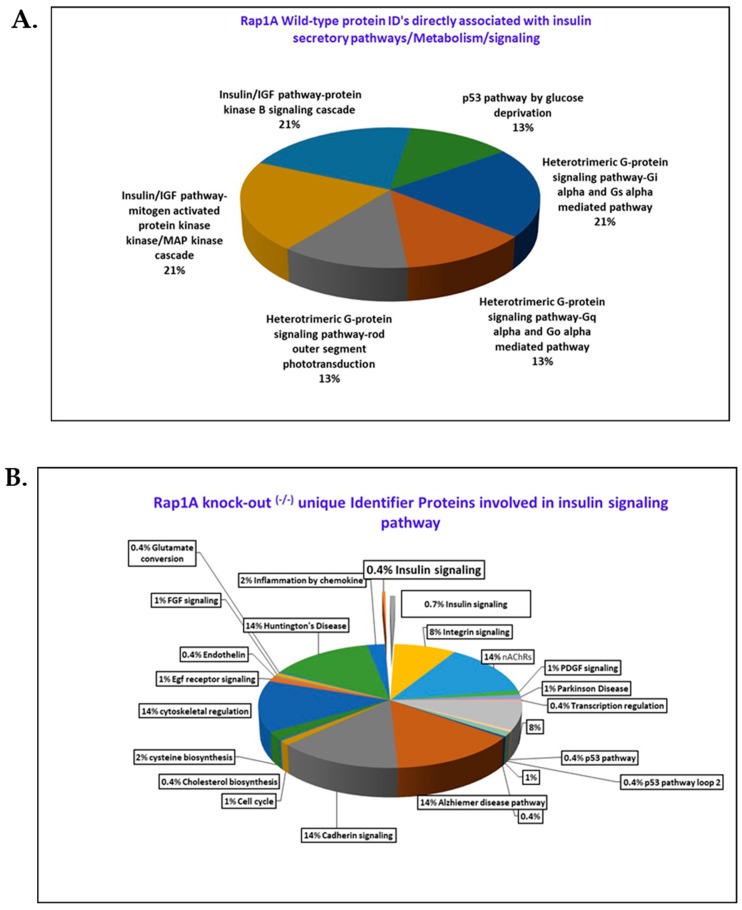
(**A**) Six quadrants of insulin signaling and G-protein signaling, specifically enriched to 15 *Rap1A* wild-type ^(+/+)^ insulin-specific identifier proteins. (**B**) *Rap1A* knock-out ^(−/−)^ mice pancreatic samples have two portions of 0.4% and 0.7% from the remaining portion of chart belonging to insulin signaling.

**Figure 7 ijms-25-08013-f007:**
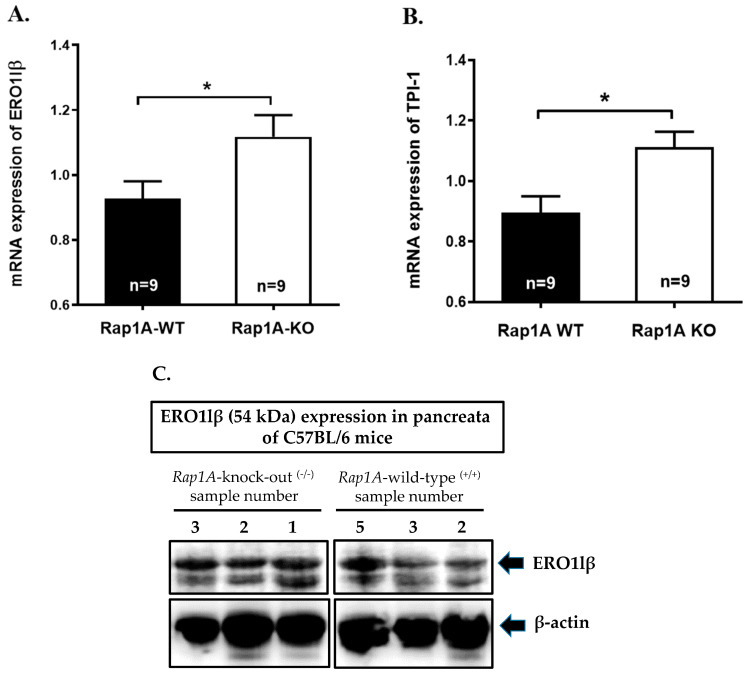
(**A**,**B**) mRNA expression assessed by qPCR for (**A**) pancreas ERO1lβ (data from n = 9 pancreatic tissues from *Rap1A* wild-type ^(+/+)^ (WT) and n = 9 from *Rap1A* knock-out ^(−/−)^ (KO) animals, * *p* < 0.05) and (**B**) pancreas TPI1 (data from n = 9 pancreatic tissues from *Rap1A* wild-type ^(+/+)^ and n = 9 from *Rap1A* knock-out ^(−/−)^ animals, * *p* < 0.05). (**C**) Pancreas ERO1lβ protein expression of *Rap1A* wild-type ^(+/+)^ and –knock-out ^(−/−)^ mice assessed by Western blot analysis. A previous study reported doublet for ERO1lβ in mouse pancreas [14]. Similarly, we observed doublets at ~50 kDa and at ~54 kDa. β-actin was used as control for loading and for data normalization (representative images). (**D**) Quantification of ERO1lβ expression (the ~54 kDa band) normalized to β-actin, expressed as mean ± SEM ratio of ERO1lβ: β-actin for *Rap1A* wild-type ^(+/+)^ (n = 5) and knock-out ^(−/−)^ (n = 5) samples. ** *p* < 0.01. (**E**) Quantification of β-actin expression for *Rap1A* wild-type ^(+/+)^ (n = 5) and knock-out ^(−/−)^ (n = 5) samples; ns, not significant.

**Figure 8 ijms-25-08013-f008:**
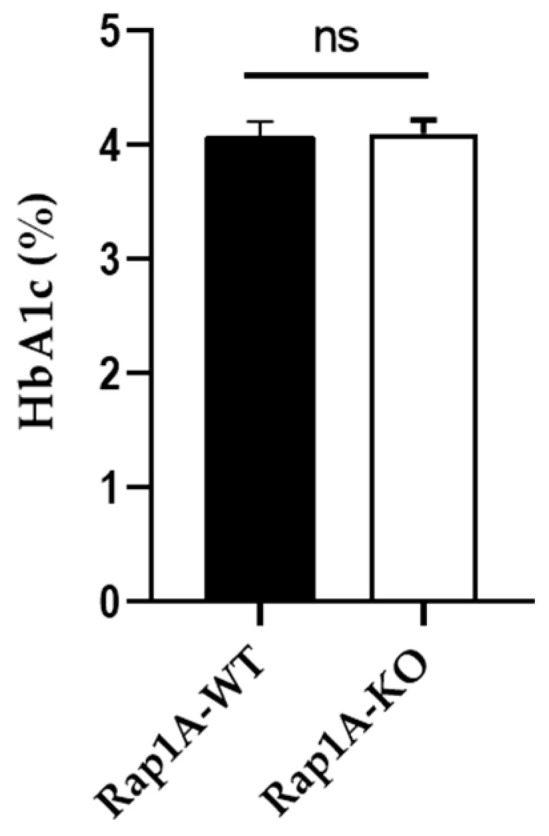
Glycated hemoglobin (HbAIc) in *Rap1A* wild-type ^(+/+)^ (WT, n = 6) and knock-out ^(−/−)^ (KO, n = 6) C57BL/6 mice; *p* = ns, not significant.

**Figure 9 ijms-25-08013-f009:**
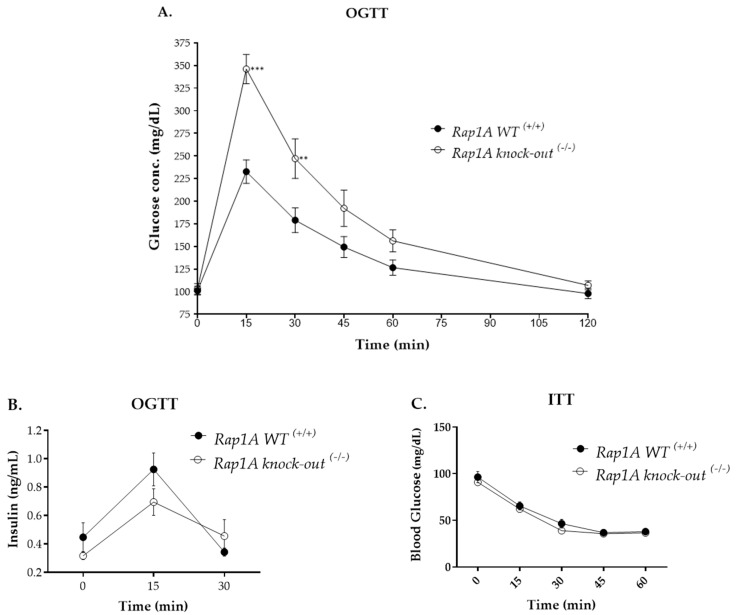
Oral glucose tolerance and insulin tolerance tests. (**A**,**B**) Oral glucose tolerance test. (**A**) Blood glucose levels of *Rap1A* wild-type ^(+/+)^ (control, black circle; number of replicates, n = 12) and knock-out ^(−/−)^ (white circle; n = 8) mice at various intervals (time, min) (**, *p* < 0.01; ***, *p* < 0.001). (**B**) Insulin levels in plasma of *Rap1A* wild-type ^(+/+)^ (control, black circle; n = 6–7), and knock-out ^(−/−)^ (white circle; n = 4–5) mice at 0, 15, and 30 min of glucose administration (*p* > 0.05). (**C**) Insulin tolerance test (ITT). Base-line corrected blood glucose values of *Rap1A* wild-type ^(+/+)^ (control, black circle; n = 12 replicates) and knock-out ^(−/−)^ (white circle; n = 9 replicates) mice at various intervals (time, min) (*p* > 0.05).

**Figure 10 ijms-25-08013-f010:**
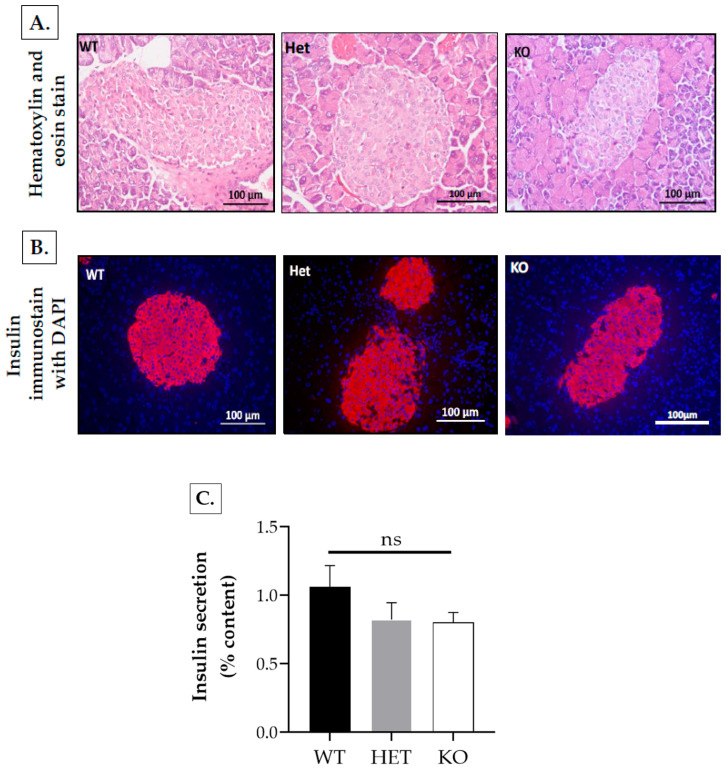
Histology of pancreatic tissue. (**A**) Pancreatic histology of hematoxylin and eosin-stained sections of *Rap1A* control ^(+/+)^ (wild-type, WT), heterozygous ^(−/+)^ (Het), and knock-out ^(−/−)^ (KO) mice. (**B**) Pancreatic immunofluorescence showing insulin and DAPI-stained β-cells of *Rap1A* control ^(+/+)^ (wild-type), *Rap1A* heterozygous ^(−/+)^ (Het), and *Rap1A* knock-out ^(−/−)^ (KO) islets. (**C**) Baseline insulin secretion from isolated islets of Langerhans in C57BL/6 mice, calculated from total insulin content, was comparatively decreased in *Rap1A* knock-out ^(−/−)^ (0.803 ± 0.071% insulin content) and heterozygous ^(−/+)^ (0.823 ± 0.122% insulin content) islets compared to wild-type ^(+/+)^ islets (1.061 ± 0.16% insulin content); however, the difference was not significant (ns, *p >* 0.05). Results are expressed as mean ± SEM. (**D**) Glucose-stimulated insulin responses of islets isolated from *Rap1A* wild-type ^(+/+)^, heterozygous ^(−/+)^, knock-out ^(−/−)^ mice upon treatment with inhibitors of protein kinase A (H-89) and Epac2 (MAY-0132). Insulin secretion from fresh isolated size-matched islets from mice of all genotypes after incubation (37 °C, 1 h) in basal glucose concentration (3 mM) and stimulatory glucose concentration (17 mM) without or with H-89 (15 µM) and/or MAY-0132 (10 µM). Actual secreted insulin contents across all genotypes were in (ng/islet/hr). Normalization of secreted insulin was performed based on size matching of islets followed by total insulin content. The figure shows results of three independent experiments, each with triplicate batches of size-matched islets/mice and each having 3 islets isolated from 2–3 male mice of each genotype. *, *p <* 0.05; **, *p <* 0.01; ***, *p <* 0.001; *p* = ns, not significant.

**Table 1 ijms-25-08013-t001:** Complete Blood Count.

Genotype		*Rap1A* Knock-Out *^(−/−)^*	*Rap1A* WT *^(+/+)^*
N=		6	5
Hemoglobin (g/dL)		11.25 ± 0.3981	10.4 ± 0.5908
Red Blood Cell Profile	Red Blood Cell Count (million/mL)	8.043 ± 0.2714	7.374 ± 0.4701
	Mean Cell Volume (MCV), (fL)	47.03 ± 0.65	48.12 ± 2.123
	Mean Cell Hemoglobin Concentration (MCHC), (G/dL)	29.73 ± 0.4240	29.50 ± 0.906
Total Leucocyte Count (10³/µL)		2.050 ± 0.7361	2.380 ± 1.122
Differential White Blood Cell Count	Neutrophils	16.833 ± 9.816	17.8 ± 12.82
	Lymphocytes	79.333 ± 10.26	79 ± 12.759
	Monocytes	2.16 ± 0.47	2 ± 0.31
	Eosinophils	1.667 ± 0.33	1.2 ± 0.2
Platelets (10³/µL)		608.7 ± 161	751.4 ± 276.8

**Table 2 ijms-25-08013-t002:** Primer sequences for genes with their respective NCBI accession number and product size.

Gene	NCBI Accession ID	Primer sequence (5’→3’) Forward (F) and Reverse (R) Primer	PCR Product Length
*Actb*	*NM_007393.5*	F-AAGTGTGACGTTGACATCCGTAAAG	307 bp
		R-TGTAAAACGCAGCTCAGTAACAGTC	
*Ero1b*	*NM_026184.2*	F-CAGGGTTTAGGAACTGCCTTG	258 bp
		R-CCAGTGTCCAAGGCTAAAAGG	
*Tpi1*	*NM_009415.3*	F-AGCACCCGGATCATTTATGG	364 bp
		R-CCACCTTGGTGACAGTTGAT	

## Data Availability

All data supporting this study are included within the article and supporting materials (see Appendix A). The mass spectrometry proteomics data have been deposited in the ProteomeXchange Consortium via the PRIDE [48,49] partner repository with the dataset identifier PXD052896.

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
