# Peer review of "Proteomic Analysis of Rap1A GTPase Signaling-Deficient C57BL/6 Mouse Pancreas and Functional Studies Identify an Essential Role of Rap1A in Pancreas Physiology"

_ijms, 2024, doi:10.3390/ijms25158013_

Round 1

Reviewer 1 Report (New Reviewer)

Comments and Suggestions for Authors

Pancreas of wild-type and Rap1A-deficient mice were investigated using proteomic and genomic methods to determine the interacting partners of the Rap1A protein, and thus the protein's function.

Unfortunately, the proteomic method used is quite outdated. Today, gel-based quantitative proteomics is not used almost anywhere, especially not on 1D SDS-PAGE. Instead, proteins isolated by immunoprecipitation are usually examined by LC-MSMS analysis.

Five mice from each groups were sacrificed for the proteomics studies, but the groups were pooled, thus leveling the individual differences only technical replicates remained. They run the SDS-PAGE and performed densitometric analysis on the CBB -stained gel. Some gel bands were in-gel digested and the peptides analyzed by LC-MSMS on a QTOF mass spectrometer. It is said that 2 µg of protein digest was injected for the analysis (l 648). One lane on a gel contained only 8 µg of proteins (l 643), so there are questions/notes emerged: -2 µg is too much for the 75 µm ID column, -did they estimate the peptide content after digestion?

What was used in the end for the semi-quantitative estimation (line 130-131)?

In the main text, there are large tables (Table 1 and 2) with the uniquely identified proteins, which is unnecessary here, in my opinion, it should be moved to the supplementary tables.

In the main text, there are large tables with the uniquely identified proteins, which is unnecessary here, in my opinion, it should be moved to the supplementary tables-

Figure 4 is a very strange, unusual and incorrect representation of the identified proteins. It should be replaced with the usual Venn-diagram.

It is claimed (line44, 882) that the mass spectrometry data has been uploaded to the ProteomeXchange repository, but at the time of review the data is not available.

Author Response

Reviewer 2 Report (New Reviewer)

Comments and Suggestions for Authors

The manuscript by Shahwar and coworkers investigates the impact of Rap1A signalling in pancreatic islets using proteomic approaches coupled with functional studies. They searched the proteomic profile of Rap1A KO mice revealing a critical role of the Rap1A signalling in insulin biogenesis, metabolism and secretion. Proteomic data were supported by in vivo and ex vivo experiments focused on insulin secretion and resistance. This study might be useful to shed light on the Rap1A interacting partners and to discover new potential targets involved in the control of GSIS.

The manuscript is well written, and the conclusions drawn by the authors are generally supported by the data. However, I have some considerations about the paper organization and some suggestions that the authors should consider to improve their manuscript.

Major comments

·     The authors should better clarify the results obtained in each section in order to make the results more sound. It would be better to add a short sentence at the end of each paragraph to point out the significance of the results.

·       The authors should reduce the number of figures in order to avoid confusion. For example, figures 8-9, figures 10-11 and figures 15-16 could be unified; it could be optimal to combine figures according to the paragraphs (i.e. figures 8-9 become one figure for the paragraph “Gene ontology Rap1A KO and WT groups: in terms of biological processes”; the same should be done for the figures above mentioned). Furthermore, it could be useful to combine figure 4 and 5 and transfer figures 6 and 7 in the supplementary section.

·       Immunohistochemistry, immunofluorescence and ex vivo GSIS were performed on WT, HET and KO mice, while the proteomic analyses were conducted only on WT and KO mice. The authors should explain the reasons of this choice in the main text.

·       In figure 15, authors showed the pancreatic islet organisation and morphology suggesting that no differences were detected between the experimental groups. Looking at the images, I noted that islets in KO mice were smaller compared to the other experimental groups. Have the authors measured the islet area? It could be important to exclude that alteration of the Rap1A signalling pathway impacts on the islet morphology, since the authors reported that Rap1A controls cell-cell contacts, migration and gap junctions which could affect islets organisation.

·       The authors described the GSIS in isolated islets in the presence of IBMX and forskolin (lines 419-439), but experimental results are not reported in the main text. Are these studies previously published? It would be important to clarify this point.

·       In Figure 16B, the authors demonstrated a significant reduction of GSIS in presence of 17 mM glucose and H-89 in KO mice, while no differences were detected after MAY-0132 treatment. Have the authors measured the basal insulin release (3 mM glucose) in the presence of the two inhibitors?

·       The authors pointed out a critical role of the Rip1A-ERO1β axis in insulin biogenesis (pro-insulin folding and insulin maturation) in the discussion. It would be insightful to measure the proinsulin levels in the isolated islets from WT, HET and KO mice to strengthen their hypothesis.  

·       Some technical details should be better clarified in the materials and methods section. For example, there is a lack of details regarding the number of islets employed for the GSIS experiments and the time between the islet isolation and the experiments. Additionally, mg of pancreatic tissue employed for RT-qPCR experiments is not reported.  Providing this information is essential for replicating the experiments and comprehensively assessing the study findings.

Minor comments

·       Antibodies catalog number should be added in the material and methods section.

·       Lines 148, 255, 632,647 and 655. The “:” should be removed at the end of the subtitles.

·       Line 797. Immunohistochemistry should be replaced by immunofluorescence.

Comments on the Quality of English Language

The English language is fine and the manuscript is overall well written. Some improvements could be made to better clarify the novelty of the paper (especially in the "Results" section as reported in "comments and suggestions for the authors").

Round 2

Reviewer 1 Report (New Reviewer)

Comments and Suggestions for Authors

The manuscript has improved a lot.

Reviewer 2 Report (New Reviewer)

Comments and Suggestions for Authors

The authors answered to all comments and revised the text accordingly.

This manuscript is a resubmission of an earlier submission. The following is a list of the peer review reports and author responses from that submission.

Round 1

Reviewer 1 Report

Comments and Suggestions for Authors

The authors of the work entitled “Proteomic Analysis of Rap1A GTPase Signaling-Deficient 2 C57BL/6 Mouse Pancreas Identifies an Essential Role of Rap1A 3 in Pancreas Physiology” develop a proteomic approach in order to elucidate the role of Rap1A GTPase in pancreas. It is a really interesting topic but, for this work, I would like to make a series of comments.

First of all, I would like to point out some aspects related to the format:

1. There are line breaks that should not be there (lines 8, 11, 41).

2. There is no homogeneity in the way of writing numbers and units. That is, in the work there is, for example, the 9µl format (without space and with the liters in lower case) and 9 µL (with space and with the liters in capital letters). Please homogenize, preferably with a space between the number and its unit and with L or l, but always the same. With respect to spaces, the same happens with ºC. Also review, please.

3. The captions of figures 8 and 9 are displaced. Please fix it.

4. Line 202 is cut by figure 10. Please fix it.

5. Why do the authors indicate, for example, 5 individuals as 05? Why the 0 before the number? Please review.

Next, I would like to point out some aspects about the methodology.

6. The authors indicate, in 4.1., that they use, in total, 5 samples from the wild strain and 5 from the mutant strain (line 303). In 4.2. they use those 5, but then in 4.6. indicate that they use 9. Could the authors clarify how many specimens did they use?

7. What mass spectrometer is used? What database?

8. Raw files, as well as search files, must be deposited in a public repository.

9. Did the authors prove that actin maintained stable expression between the two conditions to be compared?

10. What does “S.no.” in tables 1 and 2?

11. What does “as unique as they appear -uniquely- only” mean in tables 1 and 2? Do they mean proteins exclusive to one of the two conditions (WT or mutant)?

Finally, some aspects about the experimental design. If I have understood correctly, the authors use 10 samples (5+5 per group) for proteomics, which is what I find especially difficult to understand.

12. As indicated in 4.2., the authors extract the proteins from the 5 samples, quantify them, run a 1D gel, do the densitometric analysis and cut out the bands that they consider important to then do an in-gel digestion and, finally, the LC-MS/MS analysis. I would understand that they made the pool, giving rise to 3 exactly equal replicas, after quantifying and before running the 1D gel, right? That would explain figures 1 and 2. After that, they would carry out tryptic digestion of the selected bands. If so, please clarify since line 319 indicates that the bands are cut before talking about the formation of the identical triplicate (line 320).

13. Did you calculate the fold change with the average signal intensity in the densitometric analysis? Why not compare one type of sample (WT) vs the other (mutant), which is how it is usually done and makes more sense? I strongly recommend that if you did not stipulate the FC by comparing samples, rather than with means, that it be carried out.

14. In Figure 2, the authors say “protein bands selection for densitometric analysis.” For. Could they mean that they were the bands that they found differentially represented?

15. If the authors wanted to make a differential expression, the experimental approach seems somewhat strange to me. Why not do a shotgun analysis of the 5 samples (or the pool with three replicates) instead of the 1D band analysis? However, it could be a valid approach if the different points being discussed are clarified.

16. I find it extremely strange that the authors only identify one protein per band (if “S.no.” means band number). This “S.no.” reaches a value of 56 in Table 2. If it is not the number of bands, what was analyzed to have that protein list? If “S.no.” means band number, I find it even more strange not to have carried out a shotgun approach since many more LC-MS/MS analyzes were needed. It may not be clear to me because of what was mentioned above, in section 4.2. I don't know at what point the samples are mixed or what is actually analyzed by MS/MS. Based on the results shown, it seems that the WT was analyzed on the one hand and the mutant on the other, which is logical, but how many replicates of each? what about the three exactly identical replicas? As I say, I find it very confusing.

17. Then the authors do analyzes focused on global analyzes with an extremely low number of proteins as input, so concluding global results could be extremely risky.

18. Another point that confuses me is the definition of “differentially expressed.” In 2.6. The authors say that in the mutant there are 8 differentially expressed proteins and 57 in the wild type. What do you understand by differentially expressed? comparing with what? A protein can be differentially expressed by being over-represented or under-represented, so I don't understand this part either.

19. Why did the authors use only the proteins that overlapped between the two groups for the PPI analysis? (2.6). By introducing only overlapping proteins, negative interactions could be lost, for example (that is, the presence of one protein in one condition inhibits the expression of another, which is no longer seen in the other condition). I think that a more complete approach would be to use all the proteins that are significantly differentially expressed (including those specific to each condition), regardless of whether they are over-represented or under-represented and, after doing the analysis, investigate the interactions between them to be able to see different types of these.

20. Likewise, I do not understand why, as I understand it, in 2.7 they use the proteins exclusive to one condition and the other, but not those that overlap with a significant differential expression. Likewise, enrichment analyzes can also give you information on which pathways can be down-represented, which is also of clear interest, especially when regulators such as RapA1 are studied.

21. In 2.8. The same thing happens, what proteins did they use as input? In this case it would be especially important to be perfectly clear since the two proteins appear that are later validated by qRT-PCR (and where, by the way, it is seen that they are not expressed exclusively in either of the two conditions, which is related with a comment that I will make below).

22. Why is the expression of only two proteins validated? Functional validations are extremely important in omics studies, so I would recommend including more proteins to be validated.

23. In line 289 the authors say that ERO1β is found only in Rap1A when, on the other hand, in figure 12 it can be clearly observed that they are not expressed exclusively in any of the two conditions, but that there is a significant difference at the level of transcribed. Even though i am aware that there are post-transcriptional regulation mechanisms, i would me more inclined to think that the absence of evidence is not the evidence of absence, especially in approaches such as proteomics. This highlights, even more, the importance of functional validations. Do the authors believe that it would have been identified with a greater number of replicates and/or using a more sensitive mass spectrometer?

24. What statistical analysis was applied for the proteomic analysis? It is specified for qRT-PCR analysis, but not for proteomics and, as I said before, I have doubts about how this analysis was performed, which is why I find it especially difficult to validate the results.

Reviewer 2 Report

Comments and Suggestions for Authors

The paper from Shahwar et al. describes a proteomics approach to characterize partners of Rap1A by comparing differentially expressed proteins from wild type and Rap1A null mice. Although the reported data are interesting there is the fundamental questions regarding physiological characterization of the Rap1A-null animal in term of glucose responsiveness and insulin secretion. Without those data this paper doesn’t provide meaningful information.

Considering the lack of these data this reviewer doesn’t recommend the publication of the manuscript.

Reviewer 3 Report

Comments and Suggestions for Authors

Comments to the author

The manuscript entitled " Proteomic analysis of Rap1A GTPase signaling-deficient C57BL/6 mouse pancreas identifies an essential role of Rap1A in pancreas physiology" has been reviewed.

This paper has attempted to identify the essential role of Rap1A GTPase in the physiological function of the pancreas. The paper is carefully written. However, what is the most important point that the authors want to emphasize in the study of Rap1A GTPase? What is the goal of the experimental design?Rap1A GTPase has too many functions; loss of Rap1A GTPase may weaken many functions in pancreatic physiology, but it does not seem to be a big problem. The writing style of the paper is also problematic: the Discussion section is a list of what is written in the Introduction and Results, and again it is difficult to understand what is emphasized in this section.

Comments on the Quality of English Language

none